# Atomic Force Microscopy Micro-Indentation Methods for Determining the Elastic Modulus of Murine Articular Cartilage

**DOI:** 10.3390/s23041835

**Published:** 2023-02-07

**Authors:** Katherine M. Arnold, Delphine Sicard, Daniel J. Tschumperlin, Jennifer J. Westendorf

**Affiliations:** 1Mayo Clinic Graduate School of Biomedical Sciences, Mayo Clinic, Rochester, MN 55905, USA; 2Department of Physiology and Biomedical Engineering, Mayo Clinic, Rochester, MN 55905, USA; 3Department of Orthopedic Surgery, Mayo Clinic, Rochester, MN 55905, USA

**Keywords:** articular cartilage, osteoarthritis, AFM, cryosectioning, chondrocytes

## Abstract

The mechanical properties of biological tissues influence their function and can predict degenerative conditions before gross histological or physiological changes are detectable. This is especially true for structural tissues such as articular cartilage, which has a primarily mechanical function that declines after injury and in the early stages of osteoarthritis. While atomic force microscopy (AFM) has been used to test the elastic modulus of articular cartilage before, there is no agreement or consistency in methodologies reported. For murine articular cartilage, methods differ in two major ways: experimental parameter selection and sample preparation. Experimental parameters that affect AFM results include indentation force and cantilever stiffness; these are dependent on the tip, sample, and instrument used. The aim of this project was to optimize these experimental parameters to measure murine articular cartilage elastic modulus by AFM micro-indentation. We first investigated the effects of experimental parameters on a control material, polydimethylsiloxane gel (PDMS), which has an elastic modulus on the same order of magnitude as articular cartilage. Experimental parameters were narrowed on this control material, and then finalized on wildtype C57BL/6J murine articular cartilage samples that were prepared with a novel technique that allows for cryosectioning of epiphyseal segments of articular cartilage and long bones without decalcification. This technique facilitates precise localization of AFM measurements on the murine articular cartilage matrix and eliminates the need to separate cartilage from underlying bone tissues, which can be challenging in murine bones because of their small size. Together, the new sample preparation method and optimized experimental parameters provide a reliable standard operating procedure to measure microscale variations in the elastic modulus of murine articular cartilage.

## 1. Introduction

The mechanical properties of a tissue affect its behavior and capabilities. This form–function relationship provides predictive and structural information about tissues in both healthy and (pre)diseased states [1,2,3]. For articular cartilage, mechanical properties and function are especially important. The primary purpose of articular cartilage is to protect, cushion, and lubricate bone ends in synovial joints. Articular cartilage comprises chondrocytes embedded in an extracellular matrix (ECM) consisting mostly of collagen type II and proteoglycans such as aggrecan [4]. This porous and negatively charged ECM retains fluids to resist compressive loads. When a compressive force is applied, e.g., during movement, some of the fluid is released to aid in joint lubrication. This biphasic nature of cartilage means that mechanical properties, particularly elastic modulus, are aggregates of matrix stiffness and fluid [5,6].

Changes in the elastic modulus of cartilage matrix predict degenerative diseases, such as osteoarthritis (OA), where articular cartilage degradation is a key hallmark. In the natural timeline of OA, the composition of the articular cartilage ECM changes before structural changes are detected. Specifically, aggrecan content decreases, while collagen concentration increases [7,8,9]. The aggrecan loss is accompanied by the dissociation of proteoglycan aggregates, changing the permeability of the matrix [10]. Normally, higher collagen content is considered favorable for cartilage; however, with OA, this change involves the replacement of type II collagen with type I collagen, which does not interact with proteoglycans efficiently and reduces the integrity and fluid content of the articular cartilage ECM as disease progresses [11]. Tracking these ECM changes would in theory allow monitoring of disease progression and therapeutic efficacy. Indeed, recent advances in the detection of human OA with magnetic resonance imaging hold great promise [12,13,14]. Reliable methods to detect articular cartilage ECM integrity in animal and in vitro models would allow for better understanding of degradation mechanisms and facilitate the testing of therapeutic agents.

Atomic force microscopy (AFM) indentation is an ex vivo method for measuring the mechanical properties of tissue at nano- and microscale. It can detect changes in cell and tissue stiffness from Pa (Pascal) to MPa. With its high sensitivity and spatial resolution, it is a powerful technique that can be used to measure cartilage changes, considering that microscale changes occur before structural damage is detected with macroscale technologies such as histology [15]. AFM can measure how OA progresses and potentially how treatments affect disease progression. Understanding how damage accumulation occurs over time will define optimal intervention timepoints and provide a means to measure the temporal and spatial effects of therapeutics on tissue quality. 

AFM indentation operates on the same principles as any compression test and is subject to many converging variables. During an AFM experiment, the tissue is compressed by a probe. The force curve (force applied as a function of the piezo displacement) is recorded. Young’s modulus E is extracted by fitting this curve with the Hertzian model [16]. Mattei et al. identified major sources of variation in tissue samples in testing that affect mechanical measurements [17]. Sample variables include fixative, storage, or subject phenotypes such as sample age or disease state [18,19]. Experimental variables depend on the indentation method chosen by the investigator, and they include the rate, the force applied, the size and shape of the probe. Subsequently, the analysis of the force curve to obtain Young’s (elastic) modulus measurements is subject to any violation of the assumptions of the mathematical model, as well as the selection of the region of the curve to be fit. For example, the Hertzian model assumes that the substrate is an infinite plane with homogenous material properties, which is an inaccurate representation of a biological tissue. Thus, comparing AFM results obtained in different laboratories can be very difficult without understanding all the variables. It also means that there could be a high degree of variation across users and their individual experimental methods, which is difficult to examine without a detailed protocol. For articular cartilage, several variations in both sample preparation and experimental method have been described [15,20,21,22,23,24,25,26,27,28,29,30], but there is little agreement between them on experimental parameters. Thus, there is no standard protocol for using AFM to measure the modulus of articular cartilage. 

Here, we reviewed the literature on articular cartilage modulus estimated at microscale using AFM micro-indentation. We then investigated how experimental parameters affect AFM measurements of a defined sample material, polydimethylsiloxane (PDMS), and murine articular cartilage. While ramp size and ramp rate had no effect on the measurements over the range investigated, indentation force and cantilever stiffness did affect the measured modulus of the PDMS gel. Our observations show that the spring constant of the cantilever affected the measured modulus the most, but with no discernable pattern. Due to this, the cantilevers that gave the most consistent results (spring constant 1 N/m) were chosen for subsequent experiments on articular cartilage samples. The amount of force applied had a limited effect on the measured modulus of cartilage in particular, and 75% of the maximum force for each cantilever was selected. This produced high-resolution data to ensure a good model fit, while also staying within the limits of our AFM and its detector. These studies identified a set of experimental parameters suitable for microscale AFM on murine articular cartilage samples.

## 2. Materials and Methods

### 2.1. Literature Review for AFM Methods on Articular Cartilage

A bibliometric review was performed in PubMed with the terms “atomic force microscopy” and “articular cartilage”. Results (n = 179) covered a wide range of applications. Studies relating to nanoscale analysis of cartilage or individual collagen fibers were excluded because that scale of analysis was beyond the scope of this project. The results were further narrowed by focusing on AFM micro-indentation. Additional literature was found by reviewing citations from the remaining 15 papers, leading to 30 articles involving articular cartilage analysis with AFM. Eighteen of these publications lacked details of AFM experimental parameters. Methodologies and protocol variables were extracted from the remaining 12 studies and compiled (Table 1).

### 2.2. AFM Micro-Indentation Modes

All AFM experiments were done in PBS at room temperature on a Bruker Catalyst Bioscope atomic force microscope mounted to an Olympus IX73 inverted optical fluorescent microscope using NanoScope 9.1 software. Sensitivity of the photodiode and estimation of the spring constant were conducted using the thermal fluctuation method on the glass slide and in PBS just prior to micro-indentation experiments [31,32]. This involved acquiring a power spectral density plot of the cantilever response under ambient conditions (i.e., room temperature and pressure, submerged in PBS). Markers were placed on either side of where the spectrum peaks above the noise, and a simple harmonic oscillator model was fit to the contained data. This allowed the software to calculate the spring constant *k* for the cantilever. This calibration was performed with every use of each probe.

To eliminate biological sample variables, preliminary AFM experiments were performed on polydimethylsiloxane (PDMS) gel with Young’s modulus close to cartilage values reported in the literature (Bruker, catalog code PDMS-SOFT-1-12M, 2.5 MPa). A total of 25 force curves were generated from indentation covering an area of 50 µm^2^. Three different areas of the PDMS substrate were analyzed for each set of AFM testing variables. For the articular cartilage experiments, the MIRO 2.0 (microscope image registration and overlay) extension through NanoScope 9.1 AFM software (Bruker) was used to combine optical pictures taken with the phase-contrast microscope. This allowed us to discriminate among cartilage ECM, chondrocytes, and subchondral bone on the tissue section. All indentations were performed in force-ramp mode, also called force spectroscopy mode, where parameters are set and individual indentations are performed in specific locations, rather than scanning mode where topographical measurements are made of the whole area. Graphs of force/z displacement are tracings of force curves made using NanoScope Analysis 2.0. Tracings were needed because curves are difficult to visualize with the interface of this software.

### 2.3. Parameter Variation on PDMS

Experimental parameters that could affect the mechanical measurements in AFM experiments include cantilever stiffness, indentation force, ramp rate, and ramp size [33]. Three different cantilever stiffnesses were tested on the basis of the variations seen in the literature: 1 N/m, 8.9 N/m, and 16 N/m. All tips were uncoated silicon cantilevers, with 5 µm borosilicate glass spherical probes (nondeformable) from Novascan (catalog code *PT.BORO.SI.5*). Specifications of the dimensions of the silicon cantilevers are provided in Appendix A. A fourth tip (silicon nitride cantilever with 5 µm borosilicate glass spherical probe, k = 0.06 N/m) was included as a comparison to the cantilevers used for soft-tissue indentation. During calibration, it became apparent that the actual cantilever stiffnesses were different from the values listed by the manufacturer. This was not unexpected, as the manufacturer (Novascan) specifies that actual cantilever stiffnesses may vary up to 50% in either direction. The 1 N/m cantilevers varied between 1 and 3 N/m, the 8.9 N/m cantilevers varied between 4 and 9 N/m, and the 16 N/m cantilevers varied between 18 and 20 N/m. In all analysis calculations, the true values of the cantilevers as determined by thermal fluctuation were used. Other parameter changes (ramp size, ramp rate, and indentation force) were made through the settings available from NanoScope.

The force applied by the AFM tip to the sample is a testing variable defined by the user, but the maximal force threshold allowed by the AFM system depends on photodiode sensitivity and the spring constant of the cantilever. This force limit was calculated for each cantilever. To study the effect of indentation force on the elasticity results, AFM experiments were performed at values representing 25%, 50%, 75%, and 100% of the maximal force. The maximum force for each probe is listed in Appendix A. For probe comparison, indentation force was kept at 50%, with ramp size at 2 µm and ramp rate at 1 Hz. At least two probes from each group (0.06 N/m, 1 N/m, 8.9 N/m, and 16 N/m) were compared. To investigate ramp size and ramp rate, probes of listed cantilever stiffness 1 N/m were used. Having selected the probe (1 N/m), ramp size (2 µm), and ramp rate (1 Hz), indentation force was varied between 25% and 100% of the maximum limit, both on PDMS and on articular cartilage. These experiments led to the final parameter selection for indentation of articular cartilage: ramp size of 2 µm, ramp rate of 1 Hz, cantilever stiffness of approximately 1 N/m, and indentation force of 75% of the safety limit. For all experiments, environmental variables (temperature, humidity, etc.) were consistent, according to the conditions inside the microscope room (room temperature and pressure).

### 2.4. Articular Cartilage Sample Preparation

All animal experiments were carried out under protocols approved by the Mayo Clinic Institutional Animal Care and Use Committee (IACUC). C57BL/6J mice were euthanized with carbon dioxide, and cervical dislocation was performed as a secondary measure before dissection. Right distal femurs were harvested, gently cleaned of soft tissue, flash-frozen in PBS-soaked gauze, and stored at −80 °C for up to 2 weeks to allow for embedding in batches. Bones were embedded in cassette molds with optimal cutting temperature compound (OCT, Tissue-Tek^®^) and frozen in cold 2-methyl-butane. Bones were oriented vertically to allow for sectioning in the transverse plane (Figure 1a). The samples were then stored at −80 °C. We note that femurs were neither decalcified nor formalin-fixed to preserve the mechanical properties of the articular cartilage. 

To overcome the challenge of sectioning bones to characterize articular cartilage stiffness, a variation on the “tape-transfer” method was developed on the basis of the “Cryo-Jane” tape transfer system and published adaptations [25,34,35,36]. Before sectioning, the samples were mounted on the specimen holder while noting the medial/lateral orientation of the bone (Figure 1a,b, Appendix A). Glass slides were prepared by placing a small piece of thin double-sided tape in the center. Scotch tape (Scotch Tape, 3M) was cut to size and placed onto the cut surface of the OCT block (glue part facing the sample surface). Cold tweezers were used to smooth it onto the surface (Figure 1d). As the handle was turned to advance the blade, the tweezers were placed below the bottom edge of the tape to ensure it did not stick to the blade (Figure 1e). As the section was cut, it remained attached to the tape in one smooth section (Figure 1f). The orientation was marked, and then the tape was placed onto the glass slide previously prepared with thin double-sided tape (Figure 1g,h). The tissue section was now on the upper surface and the tape was on the bottom surface attached to the glass slide (Figure 1f). Slides were stored at −80 °C before AFM indentation.

### 2.5. Cartilage ECM Parameter Variation and Selection

Parameters selected from PDMS experiments were confirmed on articular cartilage samples. Two consecutive slides were taken from each sample. On each of these slides, two 150 µm square areas were selected from the medial side for analysis. On each of these areas, 25 indentations were performed on the middle zone cartilage. These areas did not overlap and were selected randomly within the posterior medial condylar cartilage. Testing variables (cantilever spring constant, ramp rate, ramp size, and indentation force) were investigated one at a time to evaluate their influence on the measurement of cartilage stiffness using the AFM micro-indentation technique. 

To compare elastic moduli across the articular cartilage samples, 15 indentations were made in two areas on the medial condyle from each of the following regions: cartilage superficial zone, cartilage middle zone, cartilage deep zone, and the subchondral bone. Two cells (chondrocytes) were also examined in the same area, where 10 indentations were analyzed from each of the following regions: center of the cell, cell-matrix border, and the surrounding matrix. This resulted in a total number of 32 chondrocytes examined across all samples. AFM parameters were kept constant (as listed above: 2 µm, 1 Hz, 75% force limit) for all measurements, except for increasing the ramp size to 5 µm for the central cell measurements, as the significant decrease in elastic modulus compared to the matrix meant that the entire force curve could not be seen at 2 µm. 

### 2.6. AFM Force Curve and Statistical Analysis

Force curves were analyzed with NanoScope Analysis 2.0 software (Bruker). The model calculates Young’s modulus E from the fitting of the force curve using Equation (1) and variables indicated in Figure 2b. To remove variability introduced by the user, the contact point selected for each curve was determined by maximizing the r² value for the model fit. If this best model fit resulted in using less than 75% of the data, the curve was discarded and not used for the results, which occurred on curves with no clear contact point. This method ensured minimal introduced error during the analysis process. For all experiments on articular cartilage, Poisson’s ratio was assumed to be 0.5 [37].
(1)$$E=\frac{3\left(1-\nu^{2}\right)}{4R^{1/2}\delta^{3/2}}F,$$
where *ν* is the Poisson’ s ratio, *δ* is the indentation depth, *R* is the probe radius, and *F* is the indentation force.

The Hertzian model makes several assumptions: (1) the substrate is an infinite plane with homogenous properties; (2) the probe is infinitely stiff compared to the substrate; (3) the probe is a perfect sphere; (4) the strain is within the linear elastic range (5–10%). The dimensions of the sample compared to the probe allow for the assumption of an infinite plane without issue. Any material, particularly biological, is not truly homogeneous, but this is an unavoidable assumption. Efforts are made to choose areas that are as close to homogeneous as possible (e.g., in cartilage not too close to subchondral bone or chondrocytes). Probes used in this study were all glass spheres; with a modulus 235,000 times greater than cartilage, the assumption of infinite stiffness is reasonable [38]. For these samples, the sections were 10 µm thick, and the depth of indentation into ECM was never more than 1 µm, remaining within the linear elastic range. When indenting chondrocytes, the depth of indentation exceeded this limit due to the high cantilever stiffness compared to the low modulus of cells. Since this study was designed to examine the matrix rather than chondrocytes, this was considered acceptable. Within each area, the analyzed points were averaged to give one point per area. All results were then plotted in GraphPad Prism 9 with data presented as the mean ± standard deviation (SD), and statistical analysis was performed using nonparametric tests [39]. 

## 3. Results

### 3.1. Bibliometric Analysis of AFM Testing Variables Used to Evaluate Articular Cartilage Stiffness

Of the many techniques that have been applied to measure cartilage stiffness, AFM is unique in that it allows precise control of the experimental environment. Topographic maps of the tissue surface can be obtained with fine tips and dynamic imaging software [15,22,24,25,26,27,36,40,41,42,43,44,45,46]. Table 1 and Figure 3 summarize AFM methods applied to cartilage in 12 publications where the mechanical measures were reported. Seven of these studies were performed with mouse tissues, two with human, and one each with rabbit, bovine, and porcine tissues. Sample preparation depended on the specimen dimensions and accessibility. Reported moduli of articular cartilage were plotted against the experimental parameters used for each study (Figure 3).

In all 12 studies, the tips used for contact mode of the tissue were spherical, which allowed use of the Hertzian contact model, but diameters varied greatly. This range may be not only due to some sample variables such as mechanical difference between species and/or regions analyzed within cartilage architecture, but also a consequence of sample preparation. The AFM tip parameters such as the spring constant and the tip radius are key factors to force sensitivity and spatial resolution. While the tip radius is accounted for in the model equation, there appears to be a slight inverse effect of tip size on measured cartilage elastic modulus (r^2^ = 0.2928) (Figure 3a). Most studies were performed with spherical tips between 2 and 10 µm in diameter, with 5 µm being the most common across all species, producing modulus values of 1 to 3 MPa for articular cartilage. 

Cantilever stiffness is an important choice during AFM parameter selection because it affects the indentation force. In the 12 studies summarized in Table 1, cantilever stiffness varied over 660-fold from 0.06 to 40 N/m [20,22,23,24,25,47], with the median lying at 7.4 N/m. The variation found in the literature is illustrated in Figure 3b, where most studies used a cantilever stiffness between 1 and 15 N/m. Across this range, the modulus results obtained appear to be independent of the cantilever stiffness. A more flexible cantilever would apply less force before reaching its safety limit than one with a higher stiffness. Due to this, comparing the indentation force directly across several studies is difficult. From the 12 studies reviewed, the force varied 50-fold from 0.04 to 2 µN (Table 1). The relationship between indentation force and measured modulus are shown in Figure 3c. As with the other parameters, there is no clear correlation. For most cantilevers of stiffness around 1–10 N/m, the force applied is around 0.2–1 µN.

Overall, there is no consensus on the best experimental parameters for AFM from the literature. Therefore, we systematically tested several variables on a control material, PDMS, followed by articular cartilage. Experimental variables tested included ramp rate, ramp size, cantilever stiffness, and indentation force. The ranges tested for each variable are listed in Appendix A.

### 3.2. AFM Variables on PDMS Control Material

The manufacturer-designated Young’s modulus of the PDMS gel was 2.5 MPa, within the range of the literature-reported modulus for articular cartilage (Table 1). Each experimental parameter (ramp rate, ramp size, cantilever stiffness, and indentation force) was varied independently of the others. Areas on the PDMS gel were not exactly matched day to day with the various cantilevers given the lack of visibility from the gel mount. However, the areas were always in the central region of the gel to avoid any edge effects.

Holding the ramp rate at 1 Hz, ramp size at 2 µm, and indentation force at 50%, probes of different cantilever stiffness were compared. The values of cantilever stiffness examined were chosen according to the literature (Table 1) and availability from the manufacturer (Novascan). The probes tested had four listed cantilever stiffnesses: 0.1 N/m, 1 N/m, 8.9 N/m, and 16 N/m. When performing thermal tune calibration, it became evident that the actual cantilever stiffnesses varied significantly from their listed value, particularly for the tips with a higher spring constant (Figure 4). Since calibration was performed at the start of every experiment, the actual cantilever stiffness was used to compare probes. There were clear differences across the four cantilever probes tested, but generally not much variation for each tip across the three areas tested (Figure 4a–d). High cantilever stiffness probes were difficult to calibrate via thermal tune, which could have introduced some degree of error. The tip with the lowest cantilever stiffness (k = 0.0891 N/m) produced very little recorded data before the force threshold was reached and was eliminated for further studies on this gel and articular cartilage. There was no pattern or trend in the relationship between the remaining cantilever stiffnesses and measured stiffnesses (Figure 4e). However, the group of 1 N/m probes was most reproducible (Figure 4e) and, therefore, selected for subsequent experiments. This group included probes of actual cantilever stiffness 1.019 N/m, 1.78 N/m, and 2.55 N/m.

With this group of probes (1 N/m), ramp rate and ramp size were kept at 1 Hz and 2 µm, respectively, while indentation force was investigated. Increased indentation force gave lower measured stiffness, and this trend followed for all three tips in the group (Figure 4f). The higher indentation forces also improved data resolution because there are more data points available for fitting with the Hertz model. On this basis, 100% of the maximum appliable force would give the best results; however, this maximum force was determined by the limit of the detector, and 75% of the force was chosen to ensure the measurements remained within the detectable region (Appendix A). This force was recalculated when each probe was calibrated.

Maintaining the parameters selected (cantilever 1 N/m, indentation force 75%), with a ramp rate of 1 Hz, the ramp size was altered. Changing the ramp size does not change the force curve (and therefore modulus) but does affect the data resolution (Figure 5a–d). Nevertheless, the force curve has the same number of data points (i.e., 512 recorded measurements, the maximum possible with Bruker NanoScope 9.1 software). Thus, a larger ramp size reduces force curve resolution and compresses useful data into a fraction of the measurement region (Figure 5c,d), making fitting of the Hertzian model very challenging and causing a large degree of error in Young’s modulus measurements. Through this reasoning, the suitable ramp sizes for the gel would be 1 µm or 2 µm. The indentations on the PDMS gel had clear sharp contact points (Figure 5a–d), but this is less likely for biological tissue, where having a larger region of low/zero signal would allow for easier baseline calibration (Appendix A). Therefore, a ramp size of 2 µm was selected. 

Finally, with cantilever stiffness of 1 N/m, indentation force of 75%, and ramp size of 2 µm, ramp rate was varied within the limits of the AFM system (0.5, 1, and 1.5 Hz). Ramp rate affects the time of indentation and, therefore, measures the viscoelastic properties of a tissue. In this case, the ramp rate did not have a significant effect on the Young’s modulus for any of the cantilever probes (Figure 5e). Ultimately, the middle setting of 1 Hz was selected as the ramp rate.

In summary, using a PDMS substrate with mechanical properties close to articular cartilage, the most suitable AFM experimental variables were determined with a spring constant k at 1 N/m, applying an indentation force representing 75% of the force limit determined by the AFM system, and force curves were established with a ramp size of 2 μm and a ramp rate of 1 Hz (Figure 6a). Next, we used these parameters to test the elastic modulus of articular cartilage samples.

### 3.3. Experimental Variables for Characterization of Articular Cartilage Stiffness by AFM

Posterior medial condylar articular cartilage on transverse sections of distal femurs from two 12-week-old female C57BL/6J mice were indented with the 1 N/m cantilevers (Figure 6b,c). One indented area of the extracellular matrix in the middle zone of articular cartilage, the cantilever tip, and examples of indented locations are shown in Figure 6c. Notably, the differences between cells and extracellular matrix are discernable through the microscope, with clear areas of matrix visible between the cells. Ramp size and rate were compared between the articular cartilage and the gel (Figure 6f–i vs. Figure 4a–d; Appendix A). The adjacent tape was also indented, to ensure that it was not affecting the cartilage measurements (Appendix A). The tape had an elastic modulus of approximately 0.2 MPa, 10 times less stiff than cartilage. Given that the sections were 10 μm thick, and that the indentations reached a depth of less than 1 μm, it is unlikely that the tape had any effect on the results.

Next, the indentation force was varied between 30 and 120 nN (representing 25% and 100% of the maximum force appliable) with a 1 N/m cantilever probe. There was a slight increase in measured Young’s modulus with indentation force on murine articular cartilage, the inverse from what was seen on PDMS gel. (Figure 4f and Figure 6d). Aside from this, there was little difference in the results seen between PDMS and articular cartilage samples; ramp rate and size did not change the results on either substrate. With these parameters, the matrix stiffness of the middle zone of murine articular cartilage was approximately 2 MPa, which is similar to some of the values obtained from the literature [15,24,25,26,27,48]. These experiments, initially investigated on PDMS gel and then confirmed on articular cartilage, established experimental parameters to be used for consistent AFM characterization of the extracellular matrix in the middle zone of articular cartilage. 

### 3.4. Spatial Stiffness across in C57BL/6J Adult Murine Articular Cartilage

Next, we used the selected AFM parameters to examine heterogeneity in murine articular cartilage samples (n = 4 male mice). The medial condyles on the right femurs from wildtype male 12 week old C57Bl/6 mice were measured. Articular cartilage is divided into three zones (deep, middle, and superficial) as a function of ECM composition and organization, and chondrocyte morphology, number, and organization [4]. As predicted, the elastic modulus of the articular cartilage matrix increased with proximity to the subchondral bone and decreased toward the superficial zone (Figure 7a,b,e). The middle zone, the largest, was around 2 MPa as found in other baseline experiments. The superficial zone was closer to 1 MPa, and the deep zone was approximately 4 MPa. Measurements on the subchondral bone tissue averaged 10 MPa; however, these numbers are imprecise and highly variable because this method was optimized for indentation of cartilage rather than mineralized bone tissue. Another method would have to be developed to gain more accurate results on bone. A similar issue occurred when indenting chondrocytes. The center of the chondrocyte was approximately 0.2 MPa, with the elastic modulus increasing around the border until reaching the surrounding matrix at again 2 MPa (Figure 7c,d,f). The spatial resolution (5 µm probe) meant that it was difficult to obtain repeated measurements of a single cell without overlapping the same area. This means that the data for the center of cells were also not accurate with these methods.

### 3.5. Articular Cartilage Matrix Elastic Modulus Is Similar in Male and Female C57BL/6J Mice at 12 Weeks of Age

Having established an experimental protocol for measurement of articular cartilage Young’s modulus, we applied it to femurs from healthy male and female 12 week old C57BL/6J mice (n = 4 female and 6 male mice per group and 8/12 femurs per group) to examine sex differences. On each section, four areas were indented on the femoral condyles, two areas each in the medial and lateral condyles (Figure 8a). Examples of indented locations in one area are shown in Figure 8b,c. The average modulus of articular cartilage was measured at 2 MPa, and the standard deviation was approximately 0.1. There was no significant difference between male and female samples, or medial and lateral areas (Figure 8d,e). These results provide a valuable baseline for healthy adult murine articular cartilage.

## 4. Discussion

Considering that the AFM technique was developed in 1986 [49] and the first experiments on biological samples analysis were performed in the mid-1990s [50], the estimation of articular cartilage stiffness using AFM micro-indentation is still rarely present in the literature. Although it has been applied to articular cartilage before [15,22,25,26,27,36,42,43,46], there is no consensus in the literature on the appropriate experimental parameters. In fact, there is great variability in experimental factors, and most studies do not justify their choice of parameters beyond literature precedent. While all methods across the literature are useful for their own application, the interdependency of experimental parameters means that they are difficult if not impossible to compare. 

We aimed to identify the optimal AFM micro-indentation parameters to measure the articular cartilage modulus at microscale. Experimental variables investigated in this study were ramp size, ramp rate, indentation force, and spring constant of the AFM tip. All variables were first tested on a PDMS gel substrate with a stiffness approximately equal to that of articular cartilage and confirmed on several regions of murine articular cartilage. The final chosen parameters were applied to male and female adult mice to determine if sexual dimorphism exists in articular cartilage elastic modulus at a healthy baseline.

One parameter not investigated in this study was tip size. For contact-mode AFM on a tissue such as cartilage, a spherical tip is frequently used. This allows the Hertzian model to be fitted to the data, producing an approximated value for Young’s modulus (E). The size of the tip is accounted for in the model (Equation (1)) and is determined largely by the scale of the region of interest. This leads to differences based on sample species and preparation method [21,22,24,25]. Since the aim of this study was to use AFM to look at the stiffness of murine articular cartilage, specifically the extracellular matrix, the probe size of 5 µm was chosen for its ability to indent the region of ECM between chondrocytes. 

Murine articular cartilage samples can be challenging to test because articular cartilage is adjacent to a mineralized tissue (bone) and cannot be removed from small bones. Soft-tissue samples are usually cryosectioned, as this allows visualization of specific areas of the tissue. Large articular cartilage samples such as those from humans or pigs can be punched or dissected away from the bone, which makes embedding and sectioning straightforward [20,42,47,51,52,53]. For murine samples, however, the cartilage is too thin to be detached from the bone. Decalcification of the bone is an option, but it takes several weeks and may affect cartilage structure. Therefore, many choose to keep the articular surface intact and sacrifice visualization of the tissue [25,26,36,43,44]. Cryosectioning with a modified tape-based method proved successful in this study, and AFM was performed on transverse sections of the distal femurs.

To eliminate uncertainty introduced by biological variability in performing AFM on articular cartilage, experimental parameters were tested on a PDMS gel model and then on murine articular cartilage samples. No parameters had significant effects on elastic modulus other than indentation force and cantilever stiffness. Increased indentation force on the gel resulted in a lower elastic modulus. This may be due to differences between PDMS mechanical properties and the assumptions of the Hertzian model. While rate-dependent behavior would explain some of these trends, ramp rate was fixed for all force experiments. Cantilever stiffness did not correlate with measured modulus but did cause it to vary wildly with different probes. This led to the selection of tips with a 1 N/m spring constant to ensure consistency across future studies. Interestingly, indentation on articular cartilage resulted in the inverse effect with force. The Hertzian model assumes the material is an infinite uniform plane being indented by a perfect sphere. While the sphere is a good approximation for the probe, the articular cartilage surface is not infinite and certainly not uniform considering the presence of chondrocytes. Comparing cartilage and PDMS directly, there is a clear difference in the shape of the indentation graph. Nevertheless, the results were very reproducible on both PDMS and cartilage with a constant indentation force. The indentation force chosen was a compromise between increasing data resolution and approaching the detector limit, which led to the choice of 75% indentation force. Using a percentage rather than an absolute force is a limitation of this study. Examining the indentation force as a percentage of the maximum allowed for comparison of many tips with dramatic differences in cantilever stiffness. Absolute indentation force values are shown to allow for interlaboratory comparison (Figure 4f, Figure 6d and Appendix A), but the final parameter choice for these studies was a percentage rather than an absolute value. In particular, the group of probes labeled 1 N/m had cantilever stiffnesses and, therefore, force limits comparatively close together. Comparing difference tips with the same absolute force would potentially result in using one tip at 100% of the limit and another at 50%. While this is a valid method, it was considered preferable to maximize the data resolution of the force curves by using a higher force when possible. Another variable not tested in this study is the Poisson’s ratio. All analyses that use a Hertzian model must assume the Poisson’s ratio to be known in advance, and many studies assumed it to have a different value, e.g., 0.1 [26], 0.3 [30], and 0.5 [37]. This parameter is included in the model and, thus, affects all results in the same way; accordingly, it was not considered necessary to investigate when focusing on experimental parameters rather than analytical.

The method was applied to adult male and female mice to determine if differences exist in cartilage stiffness at 12 weeks of age. According to measurements taken on both the medial and lateral condyles, both males and females were measured around 2 MPa. Sex differences in articular cartilage stiffness were not expected because, despite the differences in rates of osteoarthritis between men and women, there are no indications that this is due to baseline cartilage mechanics. Additionally, these studies were performed on young (12 week old) wildtype C57Bl/6 mice, which do not present with premature osteoarthritis or other potentially confounding pathology. Including this baseline allows future examination of sex differences in post-traumatic osteoarthritis. Although comparing empirical values across studies is difficult, the elastic moduli reported here are similar to the values found in the literature for small mammalian articular cartilage stiffness under similar conditions (probes on the scale of 1–10 µm), which generally varies between 0.5 and 3 MPa [15,24,25,26,43,48]. Specifically, studies on murine samples with similar parameter choices obtained stiffnesses around 1–2 MPa [26,27]. The same method was also used to examine the elastic moduli of different zones of articular cartilage and chondrocytes. The results of these experiments were as one would predict; elastic modulus increased with proximity to the subchondral bone and decreased in the superficial zone. Previously, all measurements were taken in the middle zone of the articular cartilage, and the measurements from the middle zone in these experiments were analogous to those seen before. This middle zone of the articular cartilage also produced consistent measurements (standard deviation 0.19 compared to 0.42 in the deep zone). Many studies found differences associated with distance from the articular surface, or the pericellular versus extracellular matrix [30,42,47,52]. Proximity to chondrocytes also decreased the elastic modulus. The parameters established in this study were optimized for extracellular matrix measurements, not the measurements of cells; thus, the indentations taken on cells overlapped significantly and may not have been as accurate as the measurements of the matrix. The assumptions of any mathematical model are a limitation of any study. The assumptions we made here are reasonable for the extracellular matrix of cartilage but not for cells (e.g., chondrocytes) or bone. While ECM measurements remained within small strain limits, this was not true for chondrocyte indentation due to the use of relatively stiff cantilevers. If accurate measurements of cells or subchondral bone moduli were the aim of a study, different probes and new sets of parameters would need to be established, particularly with lower cantilever stiffnesses and indentation forces.

For simplicity, we only tested one sample preparation method, cryosectioning the whole bone and articular surface. When designing this study, it was decided that the bones would ideally be sectioned to allow precise localization of the indentation measurements, which cryosectioning permitted. Since the tape-transfer method proved successful, the whole-bone method (where the articular surface is kept intact and the articular cartilage is indented blind [15,22,26,44]) was not attempted and, thus, was not evaluated in this study. 

Due to the methodological nature of this study, translation is not imminent. However, this method can provide new insights into the mechanobiology of articular cartilage both in healthy and disease states. For example, there is evidence that AFM can detect changes in cartilage before conventional methods, such as histology. This indicates that there are changes taking place in the tissue that we do not currently understand. By identifying these changes, we may be able to find new biomarkers or interventions to treat osteoarthritis in its early stages, which would improve the quality of life for millions of people.

AFM is a valuable tool for measurement of the mechanical properties of biological samples, including articular cartilage. However, the differences in how the method is applied across the literature means that no consensus method or set of experimental parameters has been established, and the extent to which parameter selection varies outcomes is unknown. As AFM has many parameters to account for, investigating their effect can enable future studies to bypass much of the methodological uncertainty. In addition, the modified tape-transfer method of sectioning non-decalcified murine bone samples ensures that even these samples, notoriously difficult to section, can be visualized on the microscope, enabling the use of MIRO and accurate localization of the indenter tip. This method is built on the basis of the Cryo-Jane system, as well as its adaptations in other studies [25,34,35,36]. Along with the experimental parameter investigation on both PDMS and cartilage samples, this method can be used to investigate murine articular cartilage under various disease and treatment conditions, which can enable valuable insight into the mechanical response of the tissue.

## Figures and Tables

**Figure 1 sensors-23-01835-f001:**
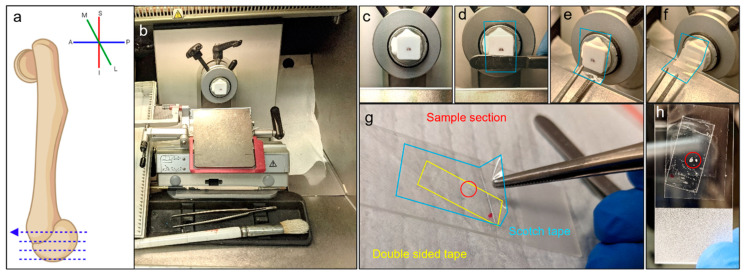
The cryosectioning process for non-decalcified murine distal femurs. (**a**) Graphic showing the orientation of the femurs during sectioning. Sections are in the lateral plane starting at the most distal part of the femur. (**b**) An image of the sectioning station at the cryostat, including mounted block, slide box, brush, and tweezers. (**c**–**f**) Process of tape transfer. The block is secured on the cryostat (**c**), and double-sided Scotch tape is smoothed onto the cut surface of the block with a tweezer handle (**d**). The tape is held with the tweezers, while the blade is advanced toward the block (**e**). The tape is pulled away with cut section on sticky side (**f**). (**g**) The tape (blue) with the section (red) is smoothed onto the prepared slide with double-sided tape (yellow) and the specimen facing upward away from the slide. (**h**) The section of the distal femur is visible on the slide after cryosectioning.

**Figure 2 sensors-23-01835-f002:**
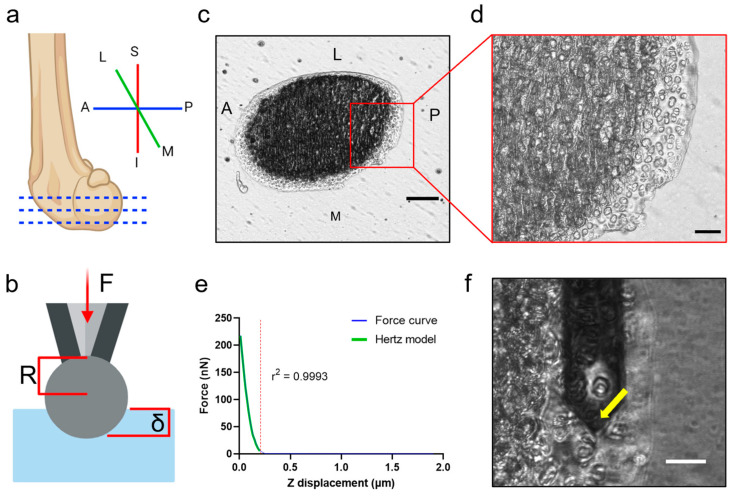
Graphics of the AFM process on articular cartilage. (**a**) Orientation of murine femora for AFM embedding. Samples are sectioned in the transverse plane, cutting from the posterior edge as indicated by dotted blue lines. (**b**) Graphic illustration of a spherical AFM probe indenting a material. F = indentation force, R = tip radius, δ = indentation depth. (**c**) A representative section of a distal femoral medial condyle. A (anterior), P (posterior), M (medial), and L (lateral). Scale bar = 200 μm. (**d**) Magnified section of articular cartilage on the surface and subchondral bone. Scale bar = 50 μm. (**e**) Indentation graphs were analyzed by Hertzian model fit. This fit covers the linear region of the curve by maximizing the r² value, including at least 80% of the data. (**f**) Image taken during AFM micro-indentation. This mode allows visualization of the sample under the microscope. Scale bar = 25 μm. The AFM tip is at the end of the yellow arrow. Note that the AFM head obscures the view of the entire sample in this setting.

**Figure 3 sensors-23-01835-f003:**
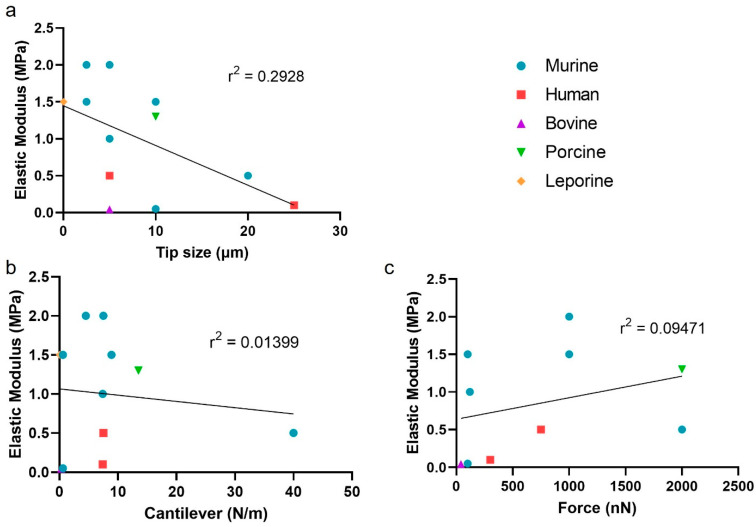
Published relationships between experimental parameters (tip size, cantilever stiffness, indentation force) and reported elastic moduli of articular cartilage. (**a**) Relationship between tip size and measured stiffness of articular cartilage. (**b**) Relationship between cantilever stiffness and measured elastic modulus. (**c**) Relationship between force and measured elastic modulus.

**Figure 4 sensors-23-01835-f004:**
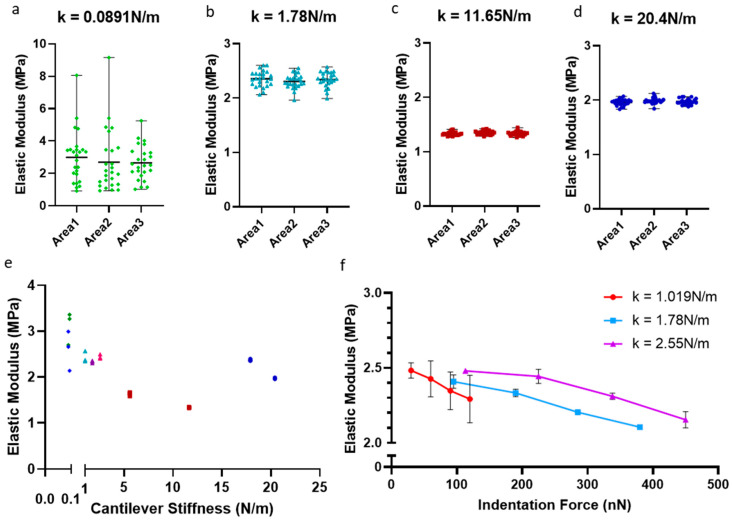
Effect of cantilever stiffness and indentation force on measured elastic modulus of PDMS gel, with a listed modulus of 2.5 MPa. Effects of cantilever stiffness and indentation force on measured elastic modulus of a PDMS gel with a listed modulus of 2.5 MPa. Areas were not the same between cantilevers, but all were in the central region of the PDMS sample. (**a**–**d**) Examples from one probe from each group with listed cantilever stiffnesses k = 0.06 ((**a**), green diamond), 1 ((**b**) light blue triangle), 8.9 ((**c**) red square), and 16 N/m ((**d**) navy dot). All indentations were performed with ramp size = 2 µm, ramp rate = 1 Hz, and indentation force 50% of maximum for each tip. (**e**) All probes were examined at 50% indentation limit; cantilever stiffness k plotted against stiffness result of PDMS gel. Probes with the same listed stiffness share a shape as in (**a**–**d**). There is no relationship between cantilever k and the result on PDMS. Probes from the k = 1 N/m group (triangles) are clustered, but others show little relationship with each other. Nominal stiffness 8.9 N/m = red squares. Nominal stiffness 16N/m blue dots. All other probes have their own color. (**f**) Effects of indentation force by three tips with listed cantilever stiffness of k = 1 N/m. Actual cantilever stiffnesses are listed in the legend. All tips were comparable, but there was a significant decrease in modulus measured with increased indentation force (see Appendix A).

**Figure 5 sensors-23-01835-f005:**
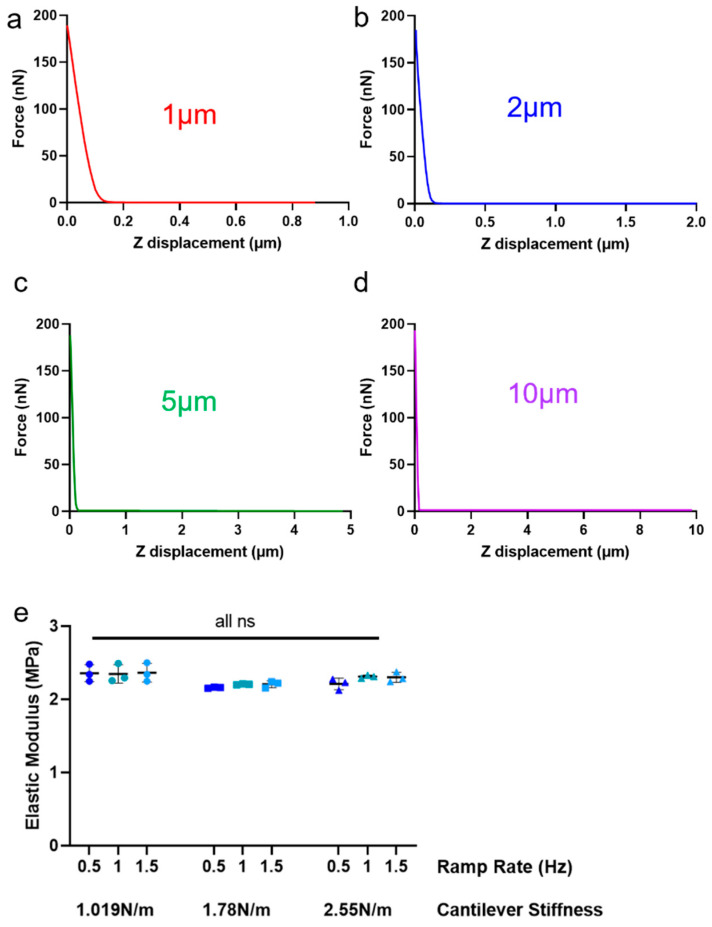
Effect of ramp size and rate on the measured elastic modulus of a PDMS gel, with a listed modulus of 2.5 MPa. (**a**–**d**) Examples of indentation curves created by four ramp sizes between 1 and 10 µm. Higher ramp sizes steepen the curve and decrease data resolution due to the large flat region preceding the useful curve. (**e**) The effect of three ramp rates on elastic moduli was determined with three different probes with cantilever stiffnesses of k = 1.78, 2.55, and 1.02 N/m. One-way ANOVA was performed and showed no significant changes with ramp rate. All indentations performed with ramp size = 2 µm and indentation force 75% of maximum (~250 nN). ns = non-significant (*p* > 0.05). Colors indicate different ramp rates; shape indicates different cantilever probes.

**Figure 6 sensors-23-01835-f006:**
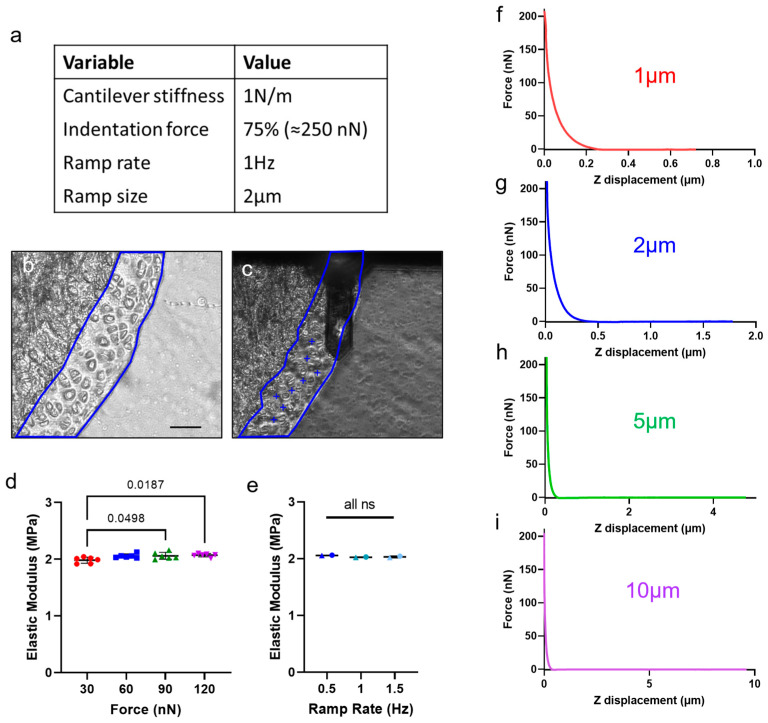
Parameter variation on murine articular cartilage. Sequential sections from medial distal femurs of two 12 week old C57BL/6J female mice were analyzed. (**a**) Area of articular cartilage to be indented (blue outline). Scale bar = 40 µm. (**b**) Same area showing example of indentation points (+) and visualization of the AFM tip and cantilever (black). (**c**) Effect of force on measured modulus. A slight increase in modulus was observed with higher indentation forces. Ramp rate = 1 Hz, ramp size = 2 µm, cantilever stiffness listed = 1 N/m. Colors indicate different indentation forces. (**d**) Effect of ramp rate on measured modulus. Ramp size = 2 µm, indentation force = 90 nN, cantilever stiffness listed = 1 N/m. Colors indicate different ramp rates. ns = non-significant (*p* > 0.05). (**e**–**h**) The effect of ramp size on articular cartilage is very similar to that on PDMS. Loss of data resolution makes large ramp sizes particularly difficult to analyze. Given the slope at the contact point, 2 µm ensures adequate “zero” data to establish baseline. Ramp rate = 1 Hz, indentation force = 90 nN, cantilever stiffness listed = 1 N/m. (**i**) Final parameters chosen for AFM of articular cartilage in future experiments.

**Figure 7 sensors-23-01835-f007:**
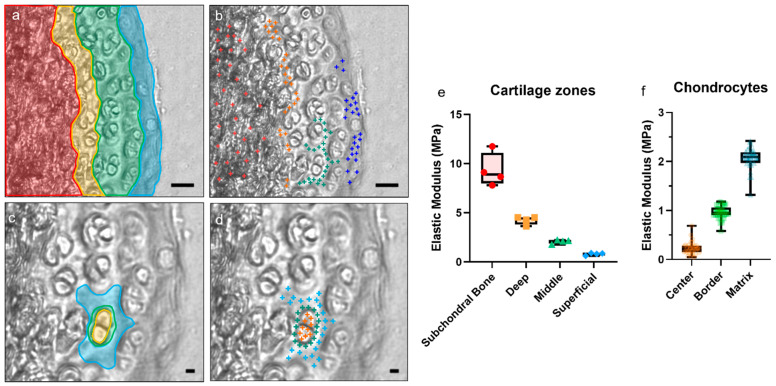
Elastic moduli vary by location in C57BL/6J murine articular cartilage. (**a**) Regions of cartilage and subchondral bone studied are pseudo-colored: Red = subchondral bone, orange = deep zone, green = middle zone, blue = superficial zone. Scale bar = 20 µm. (**b**) Actual points indented for each zone are indicated by colored plus signs. Scale bar = 20 µm. (**c**) Regions surrounding chondrocytes. Yellow = cells, green = pericellular region and cell membrane, blue = extracellular matrix. This figure shows one chondrocyte from one area; due to the proximity of some chondrocytes, adjacent chondrocytes such as those highlighted were considered a single cell. Two chondrocytes were analyzed in each area. Scale bar = 5 µm. (**d**) Actual points indented for each area surrounding the chondrocyte are indicated by plus signs. Scale bar = 5 µm. (**e**) Elastic moduli for each area of cartilage examined. Four animals were tested as biological replicates; each point represents the average for a single animal. (**f**) Elastic moduli for regions surrounding chondrocytes; each point is one cell.

**Figure 8 sensors-23-01835-f008:**
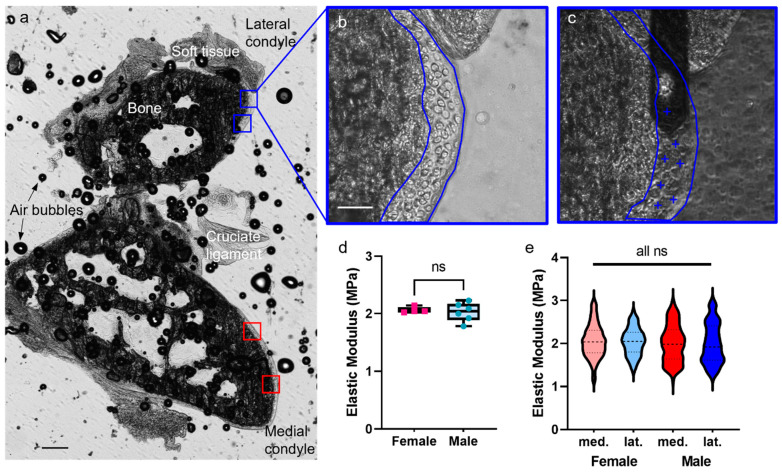
Sex does not affect elastic modulus of adult murine articular cartilage. (**a**) Section of a distal femur of a 12 week old C57BL/6J female mouse, showing four areas indented on medial and lateral condyles (red and blue boxes, respectively). A cruciate ligament is visible, as is the internal bone structure. Air bubbles between layers of tape are unavoidable; although visible, they do not affect sample section on surface. This image comprises two 4× images stitched together (aligned on PowerPoint and grouped), scale bar = 0.2 mm. (**b**) Example area of the articular cartilage on the lateral side. Scale bar = 40 µm. (**c**) An image showing indentations (blue plus signs) and visualization of AFM tip arm. (**d**) Comparison of articular cartilage elastic modulus from male (n = 6, blue circles) and female (n = 4, pink square) mice. (**e**) Violin plots of data on medial (red) and lateral (blue) articular cartilage from males and females (n = 3 each). Each point represents an average of 25 points within an area, and each sample was indented in two areas on both medial and lateral sides. Dashed lines indicate the median and quartiles. ns = non-significant (*p* > 0.05).

**Table 1 sensors-23-01835-t001:** Summary of AFM methodology in 12 studies of articular cartilage where experimental parameters were available.

References	Species	Sample Preparation	Spherical Tip Radius (µm)	Cantilever Stiffness (N/m)	Force Applied (µN)	Measured Elastic Modulus (MPa)
Christensen et al. (2012) [22]	Mouse	Whole bone glued to dish	10	0.6	0.10	**0.05**
Azadi et al. (2016) [23]	Mouse	Whole bone sanded thin	20	40	2	**0.5**
Li et al. (2015) [24]	Mouse	Whole bone—flat areas	5	7.4	0.12	**1**
Li et al. (2020) [15]	Mouse	Whole bone glued to slide	10	8.9	1	**1.5**
Chery et al. (2020) [25]	Mouse	Cryosectioned—tape transfer	2.5	0.6	0.10	**1.5**
Batista et al. (2014) [26]	Mouse	Whole bone glued to disk	2.5	4.5	N/A	**1–2**
Doyran et al., (2017) [27]	Mouse	Whole bone glued to slide	5	7.5	1	**1–2**
Danalache et al. (2019) [21]	Human	Cryosectioned from plug	25	7.4	0.30	**0.1**
Wilusz et al. (2013) [20]	Human	Cryosectioned	5	7.5	0.75	**0.5**
Tomkoria et al. (2004) [30]	Rabbit	Osteochondral punch	0.02	0.06	N/A	**1–1.5**
Park et al. (2004) [28]	Cow	Osteochondral plug	5	0.32	0.041	**0.045**
Loparic et al. (2010) [29]	Pig	Cartilage cut off bone	10	6.5–27.5	2	**1.3**

## Data Availability

Data is contained within the article or Appendix A.

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
