# Peer review of "Atomic Force Microscopy Micro-Indentation Methods for Determining the Elastic Modulus of Murine Articular Cartilage"

_sensors, 2023, doi:10.3390/s23041835_

Round 1

Reviewer 1 Report

Arnold et al. present a well-structured and adequately presented manuscript for the determination of the component of the extracellular matrix. The authors make an adequate introduction to the state of the art, and manage to justify the hypothesis and the spirit of the study. The authors manage to contribute novelty in the area of knowledge. The methodology is correct and reproducible. The authors justify the sample size precisely, and the manuscript is given strength. The results are robust and complete, and support the conclusions. The discussion is complete, however, the authors should discuss a little more the limitations of the study and the translation in the specific field. I would suggest to the authors to improve the quality of the tables and figures. Especially table 1 and figure 1.

Please review references 7, 31 and 50. There is more up-to-date evidence in this regard.

A graphic summary can help authors to better convey the ideas.

Reviewer 2 Report

In this work, the authors use AFM micro-indentation methods for determining the elastic modulus of cartilage. They propose a new sample preparation method and optimized parameters for reaching a reliable measurement. These explorations are in general helpful, although not significant.

The experimental part is quite simple, and no doubt the results are well analyzed. But there are many issues that have to be further clarify, and reviewer believe that a major revision is necessary.

It is very important to discuss the calibration method used for the cantilever stiffness k, if they want to optimize the experimental parameters. This is even more important when you have a deformable sphere on your tip apex. So, please describe the calibration in details.

Equation 1 is an approximation when the elastic moduli of glass sphere is much higher than the cartilage. This has to be clarified here.

In Figure 4, how are the three areas chosen? Are they random areas, e.g., for different cantilevers, whether the areas1-3 are the same areas on the cartilage?

The circled scatters in Figure 4e looks different to 4f. Why?

Please describe the force-ramp mode of AFM in the methods part.

Figure 5, a-d, please check the label of x-axis is really indentation or z-piezo movement.

The conclusion of indentation force and cantilever stiffness has to be enhanced. The authors report that both have effects of the elastic modulus. However, they are not really separately studied. For different cantilever stiffness, at the same relative indentation force (e.g. 75%), the absolute value of indentation force is different. There, you always get results having effects from both absolute force difference and stiffness difference. Therefore, a comparison at the same absolute indentation force is necessary.

Reviewer 3 Report

In the manuscript entitled “Atomic Force Microscopy Micro-indentation Methods for Determining the Elastic Modulus of Murine Articular Cartilage”, by Katherine M. Arnold et al., the Authors presented the nanomechanical investigation of cryosectioned epiphyseal segments of articular cartilage by Atomic Force Microscopy. The Authors started with the literature semi-systematic analysis of performed experiments. Then the study of PDMS samples was presented, on which Authors determined experiment conditions for proper measurements of the tissue. The aim of the paper was to offer experimental conditions for comparative studies between the reports. In my opinion such an approach resulted in unnecessary elongation of the paper which should rather focus on the experimental data. The material tested, sample preparation and overall biological aspect of the paper is of great interest to me and I am sure that also to the readership of the journal. However, the methodology is not well described which would hamper for reproducibility of the data lacking basic information.
In particular:
In line 149 the Authors describe loading force “values representing 25, 50, 75 and 100% of the maximal force”. However, the maximal loading force is not presented. From the discussion (line 503) I understood that it is based on the limit of the detector. Such an approach would result in hardly repetitive experiments in different laboratories, as the detector sensitivity varies between microscopes and their manufacturers.   The Authors should rather use loading force values in units Newtons, based on the proper calibration (detector sensitivity tests).
In lines 390 and 330 the Authors indicated that the resolution of the force curves is weak. 512 points can be used for very clear samples, but when changing the experimental conditions, such as ramp size, cantilever type etc. much more points should be set in the experiments. Even the Authors indicated that it might affect the accuracy of the results presented. I would advise performing the topography measurements using sharp cantilevers. If the roughness is >200 nm it would indicate that the indentation presented (usually <500 nm) will be largely determined by the topography of the sample.
The data on PDMS should be put in the supplementary information. They present no novelty and should be used by the Authors only for the purpose of their understanding of the physics of the interaction between the cantilever/tip apex and the sample. Even the Authors concluded in line 276 “Overall, there is no consensus on the best experimental parameters for AFM”. The cantilever and tip should be selected in that way that the calculated Youngs modulus should not be dependent on the indentation depth. It might be the case that the colloidal tip or cantilever is nonuniformly deformed on such stiff samples. It is difficult to interpret these data, as Authors did not provide detailed information about the cantilevers used in the experiments. There are data on spring constant, but not about the length and shape of the cantilever. The direct name of the cantilever would be helpful.
In the Figs 5 a-d and 6 e-h the presentation of data is incorrect. When reading the graphs for 0 values on the x axis, meaning indentation of 0 um the force (y axis) is maximal. Moreover, the scales between graphs are not unified which hamper easy comparison of the shapes of indentation curves. Presenting the points AND the fit of the model should be performed especially if there is uneven number of points in the indentation curves. Presenting the method for determination of contact point is an advantage that usually is lacking.
I would advise for using z range and approach retract speed instead of ramp size and rate. It is easier to follow when values change.
In Figure 7 what I would recommend is a force volume map presenting different areas and transition zones. For these type of samples – sections supplemented with optical image it would be a huge advantage. Still, the data and way of presentation of figure 7 has the biggest value of the paper.
The Authors presented male and female data. It is advantageous to present gender specific data, but I missed the discussion of the results. Are there scientific indications to expect (or not) gender differences?

Minor:
The references should be given to the articles, rather than to company brochures – ref 34, 35.

To sum up, I would advise rebuilding the paper and repeating the AFM experiments with more data points in the force curves, especially when the indentation is small – 200-300 nm. The PDMS can be placed in SI. More AFM results using for example sharp and colloidal probes could be presented. Discussion of elastic vs plastic deformation for such stiff samples would be a value. In the present form, I advise against publication.

Round 2

Reviewer 2 Report

I thank the authors for their efforts. All of my questions are solved and I think the manuscript can be accepted in its current form.

Author Response

Please see pdf attached.

Reviewer 3 Report

I would like to thank the Authors for the great improvement of the paper after the first round of review. I would recommend the publication with one remark which was not addressed properly. The Authors in the text and in the response clarify that the manuscript is (also) addressed to scientists taking their first steps in AFM measurements. In order to keep the way of presentation of AFM data as standardized as possible the report should follow the common way of presenting data. Therefore, I cannot agree to presenting the loading force in the percent of the force curve. In such a form the data cannot be easily compared between the laboratories, the results would be dependent on the cantilever, instrument and setting of the laser on the detector. That is the way why we use the calibration on a stiff substrate. In contrast to the stiffness parameter (expressed as the first derivative of the slope) elasticity parameter/Young's modulus presents the data that in assumption should not be dependent on the cantilever type. The elasticity parameter of the sample should not be highly dependent on the cantilever used and therefore this parameter, expressed in Newtons, should be used in the data presentation and comparisons. It is the only way to compare data between laboratories. Still, the differences were observed, as indicated in the big experiment ( https://doi.org/10.1038/s41598-017-05383-0 ). To sum up, if the paper should meet the standards and be used for newcomers to AFM methodology they should meet the standards of the measurements.
